# A Comprehensive Approach Limiting Extractions under General Anesthesia Could Improve Oral Health

**DOI:** 10.3390/ijerph17197336

**Published:** 2020-10-08

**Authors:** Nicolas Decerle, Pierre-Yves Cousson, Emmanuel Nicolas, Martine Hennequin

**Affiliations:** 1Université Clermont Auvergne, Centre de Recherche en Odontologie Clinique (CROC), F-63000 Clermont-Ferrand, France; nicolas.decerle@uca.fr (N.D.); p-yves.cousson@uca.fr (P.-Y.C.); emmanuel.nicolas@uca.fr (E.N.); 2CHU Clermont-Ferrand, Service d’Odontologie, F-63003 Clermont-Ferrand, France

**Keywords:** access to care, oral rehabilitation, general anesthesia, oral health related quality of life, mastication

## Abstract

Access to dental treatment could be difficult for some patients due to dental phobia or anxiety, cognitive or sensorial disabilities, systemic disorders, or social difficulties. General anesthesia (GA) was often indicated for dental surgery, and there is almost no available data on adapted procedures and materials that can be applied during GA for maintaining functional teeth on the arches and limiting oral dysfunctions. This study evaluates changes in oral health-related quality of life and mastication in a cohort of uncooperative patients treated under GA according to a comprehensive and conservative dental treatment approach. Dental status, oral health-related quality of life, chewed bolus granulometry, kinematic parameters of mastication, and food refusals were evaluated one month preoperatively (T0), and then one month (T1) and six months post-operatively (T2). One hundred and two adult patients (mean age ± SD: 32.2 ± 9.9 years; range: 18–57.7) participated in the preoperative evaluation, 87 were treated under GA of which 36 participated in the evaluation at T1 and 15 were evaluated at T2. Preoperative and postoperative data comparisons demonstrated that oral rehabilitation under GA helped increase chewing activity and oral health-related quality of life. The conditions for providing dental treatment under GA could be arranged to limit dental extractions in uncooperative patients.

## 1. Introduction

Access to dental treatment may prove difficult for some patients due to dental phobia or anxiety, cognitive or sensorial disabilities, systemic disorders, or social difficulties. Depending on the biopsychosocial situation of patients, the context of access to dental treatment may vary. For example, individuals with dental anxiety have irregular dentist attendance patterns, which are characterized by either never visiting a dentist, or visiting only when experiencing pain, while persons with learning difficulties or neuromotor disabilities do get access to dental treatment [1,2], but receive fewer dental treatments despite having greater need [3,4]. Independently of each situation, all these individuals have poorer oral health [5,6,7], more dental pain, and a poorer quality of life compared to the general population [8,9]. The use of conscious sedation was limited due to many patients’ specific needs. The relative sedative effects of the drugs limited the intervention time and required several consecutive interventions, leading to referrals for dental treatments under general anesthesia (GA) for most patients. Dental treatment under GA was well documented for children. In adults, GA was mostly indicated for dental extractions and oral surgery, and few studies reported indications for conservative care and oral rehabilitation [10,11].

For uncooperative patients, comprehensive oral rehabilitation under GA is challenging. On one side, the intervention time and reiterations should be limited in order to reduce the morbidities and mortality rate of the procedure [12]. As dental extractions require less time than endodontic treatments or restorative and prosthetic treatments, dental treatment under general anesthesia has for a long time been commonly associated with oral surgery and tooth extraction [13,14]. On the other side, extracting teeth impacts the performance of mastication, and particularly that of any patients unable to cope with a removable denture for behavioral reasons, or for whom replacements with fixed dentures or implants is not possible for financial reasons. A recent study described procedures and outcomes of restorative and endodontic treatments performed under GA and discussed these protocols with regard to the literature [15,16]. The recent developments in dental instruments, and particularly those for endodontic treatments, could assist in reducing intervention times. It was demonstrated that endodontic treatments can be provided under GA in accordance with academic standards of treatment [17].

It was reported that GA facilitates achieving dental treatment in uncooperative children [18]. However, the full mouth rehabilitation in adults is more difficult for several reasons. First, the dental treatments on permanent teeth have to survive for the entire life span of the individual, while the expected survival time on deciduous teeth is less than 10 years. Second, in most of countries, when outpatients are treated in one-day surgery, the duration of the intervention is limited to two hours. However, treating 32 teeth required a longer time than for treating 20 teeth. Third, depending on the countries, the anesthesiologists’ team, the equipment, and the clinical cases, the patient could be intubated either nasally or orally. Recording the patient bite to adjust the occlusal restoration is always difficult under GA, even when patient are intubated orally as the tube and the packing puts the tongue and the jaw forward. In case of oral intubation, it is necessary to move the tube on one side in a retromolar position in order to guide biting. Such manipulations cost additional time. For all these reasons, it is legitimate to doubt the outcome of occlusal restorative procedures performed under GA [17,19,20]. Measuring changes in the chewing ability in patients treated under GA would answer these questions.

This study aims to evaluate changes in oral health-related quality of life and mastication in a cohort of uncooperative patients treated under GA according to a comprehensive conservative approach.

## 2. Materials and Methods

### 2.1. Type of Study

This prospective follow-up study was designed to ascertain whether treatment under GA could offer a functional benefit in uncooperative patients with deficient mastication due to multiple carious lesions. The study was approved by the local ethics committee (CECIC: 2010/06; IRB number: 5044). Written informed consent was obtained from all patients. This study conformed with Strengthening the Reporting of Observational Studies in Epidemiology (STROBE) guidelines for human observational studies.

### 2.2. A Comprehensive and Conservative Approach for Dental Treatments under GA

The approach was designed to satisfy all administrative obligations for access to GA. During a preoperative consultation, patients were examined by one practitioner and an expert in special care dentistry, and they were then referred for dental treatment under GA according to the French National Authority for Health recommendations [21]. Where possible, retroalveolar and/or orthopantomogram radiographic examinations were carried out. In cases of acute dental disease, antibiotics and/or antalgics were prescribed. Patients were scheduled for a one-day surgery session and received oral and written information on the aim and conditions of the dental treatments. Patients were then referred to the anesthetist for a preoperative consultation around two weeks prior to the GA session. Any one-day surgery performed in a French hospital is limited to established, low-risk, simple, and short interventions (1–1.5 h) [22]. Significant increases in pulmonary complications were reported for surgical interventions between 2.5 and 4 h of surgical time [23,24]. These criteria may be broadened in light of the team’s experience with more prolonged interventions, provided that the duration of post-anesthetic monitoring remains compatible with the structure’s operating schedule. For patients requiring multiple dental treatments, two consecutive interventions were organized at a four to six month interval. These conditions were subject to prior agreement between the operator and the anesthetist.

On the day of surgery, patients were asked to take their current medication while fasting. Patients who demonstrated high levels of anxiety received 5 mg of midazolam orally, before going into the operating room. After anesthesia was administered, oral or nasal intubation was performed. Induction and anesthesia were administered by an anesthetist and monitored by an anesthetist nurse. An expert in special care dentistry, assisted by a resident dentist, performed the dental procedure. An operating room nurse provided the instruments and materials. The procedures for dental treatments were structured to limit teeth extractions. Conventional procedures were adapted with a twofold objective of (i) keeping functional and asymptomatic teeth on the arch and (ii) having the shortest possible duration (Table 1). At the end of the intervention, the patients were directed to the recovery room under the anesthetist’s supervision. After extubation, the patients’ cardiovascular and respiratory functions were monitored for two hours before being transferred into another room. After recovering all their functions and consuming a meal, they were escorted home by their carers or family. The next day, a nurse phoned them to check for any adverse reactions to the intervention. A one-month postoperative check-up was scheduled. For the subjects included in the cohort, one additional postoperative date was organized six months after treatment. With regards to finances, all costs for treatment under GA in a one-day surgery were covered by the French social insurance.

### 2.3. Study Population

The cohort was populated by uncooperative adult patients (between 18 and 65 years old) referred by general dentists or physicians to the Special Care Unit of the Clermont-Ferrand University Hospital after failure to be treated in the dental chair, from 01/01/2011 to 31/12/2015. Only patients with at least one decayed tooth at stage 3 or 4 on the Ekstrand Classification of caries progression on each quadrant were included [26]. Patients with learning disabilities, or those who did not speak French, were not included due to their inability to fully participate in the study data collection. The inclusion/exclusion criteria are detailed in Table 2.

### 2.4. Experimental Procedure

For each included patient, the evaluations of mastication and oral health-related quality of life (OHRQoL) were collected one month before (T0), one month after (T1), and six months after (T2) the dental treatment GA session.

### 2.5. Study Criteria

#### 2.5.1. Dental Status

Dental status was characterized using the sum of the number of decayed, missed and filled teeth (DMFT index) [27], and with the number of posterior functional dental units (PFU): PFU is defined as a pair of posterior antagonist natural teeth with at least one contact area during chewing. The Index of Dental Anxiety and Fear (IDAF 4C+) was proposed to any patients receiving GA [28]. The French version was produced after a turn/return translation process involving three native English speakers and three native French speakers. Modules 1 and 2 characterized dental anxiety (when scores are higher than 20) and dental phobia (when scores are higher than 7.5), respectively.

#### 2.5.2. Dental Treatments

The duration of the interventions, as well as the number and type of dental treatments performed, were recorded.

#### 2.5.3. Oral Health-Related Quality of Life

The French version of the Global Oral Health Assessment Index (GOHAI) was used to evaluate expected changes in oral health-related quality of life (OHRQoL) [29]. The GOHAI questionnaire comprises 12 items grouped into three dimensions: (i) the functional dimension (eating, speaking, swallowing); (ii) the psychosocial dimension (concerns, relational discomfort, appearance); (iii) the pain or discomfort dimension (drugs, gingival sensitivity, discomfort when chewing certain foods). Additive (ADD) scores for the GOHAI were obtained by totaling the response codes for the 12 items. Consequently, the ADD-GOHAI ranged from 12 to 60, with a higher score indicating a better reported oral health. An ADD-GOHAI score of 57 to 60 is regarded as high and corresponds to a “satisfactory” oral health-related quality of life. A score from 51 to 56 is regarded as “moderate” and a score of 50 or less is regarded as a “low” score, reflecting a poor OHRQoL.

#### 2.5.4. Mastication Evaluation

At each step of the experimental procedure (T0, T1, and T2), participants were asked to chew three types of test foods: soft gelatin, hard gelatin, and raw carrot. Each session began with the mastication of four pieces of gelatin (two soft and two hard) provided at random. Subjects were then asked to chew three pieces of carrot. The first piece of carrot was used to evaluate the duration of spontaneous mastication before swallowing. For the last two pieces of carrot, the participant was asked to spit each bolus when he/she was ready to swallow. When the chewing time was five seconds longer or shorter than the duration of spontaneous mastication before swallowing, the test was repeated.

Carrot bolus granulometry: Each expectorated carrot bolus was collected in a container, rinsed with water in a 100 mm sieve to eliminate saliva, and dried at 80 °C for 30 min. Each bolus was then spread on a transparent A4 sheet and scanned (Epson perfection 4990 photo™). The 600 dpi obtained images were then processed in order to evaluate food particle size and distribution (Powdershape^®^, IST AG, Switzerland), thus obtaining the D50 value. The D50 value was characterized as the median size of the carrot particles. As recorded in a previous study, the two D50 values recorded for each subject were averaged. The patients with a D50 value above 4 mm, defined as the normal threshold by the masticatory normal indicator (MNI), were considered to have impaired mastication [30].

Chewing kinematics: The evaluation of each kinematic parameter during mastication was carried out through video recording. The recorded variables were chewing time (CT: the time in seconds between the moment food is placed in the mouth and swallowing) and the number of chewing cycles (CC: number of chewing actions during the CT period). The chewing frequency (CF) was calculated as the CC/CT ratio. It was demonstrated that a lack of change in mean chewing frequency values could be used as a criterion for good masticatory health and alternatively, large variation in mean chewing frequency values could be indicative of an impaired masticatory function [31].

Food refusals: food refusals were considered each time the subjects refused to test the food sample and each time the subjects expectorated the sample before the end of the first chewing cycle.

### 2.6. Study Hypotheses

Referring to previous studies, it was hypothesized that, after treatment under GA, the patients would improve their oral health-related quality of life, particularly because of the lack of pain [32]. Considering the poor precision in procedures for coronal restorations, and the potential number of extractions, it was also hypothesized that chewing abilities would decrease after treatment, which could be confirmed by an increase in carrot bolus granulometry and differences in chewing frequencies between hard and soft gelatin.

### 2.7. Sample Size Calculation

The number of subjects required for measuring a possible difference between ADD-GOHAI values before and after rehabilitation under GA was estimated in a pilot study carried out on the first 10 patients who attended the one-month postoperative appointment. For this group, the ADD-GOHAI mean values were 38.50 ± 11.98 before GA and 52.70 ± 0.63 one month after GA. The one-way comparison of these observed mean values, performed with the epiR package 0.9-30, was based on a difference of 14.2 with a common standard deviation of 12.51 and forecast the number of subjects required at 15 (α = 5%, β = 10%).

### 2.8. Statistical Analysis

SPSS© version 20.00 (IBM SPSS statistics) was used for the statistical analysis. D50 mean values, CC, CT, CF, and ADD-GOHAI scores were compared between T0 and T1 by paired Student t-tests for patients evaluated at T1, and between T0, T1, and T2 by repeated measure procedure for patients evaluated at T2 (α = 0.05). Groups’ representativeness in terms of age, sex/ratio, oral statuses, and mastication criteria was investigated through independent intergroup comparisons or preoperative data. The data of patients evaluated only at T0 were compared to those of patients evaluated at both T0 and T1. The data of patients from the T0 and T1 evaluations were compared to that of the patients participating in the T0, T1, and T2 sessions. A Pearson correlation was applied to look for relationships between the number of PFU and the bolus granulometry.

## 3. Results

During the five-year study period, 1834 patients were treated under GA, of which 1014 were children under 18 years old, 102 were patients over 65 years old, and 367 were adult patients with learning disabilities. Finally, 351 patients were eligible for this study and 102 (45 men and 57 women, mean age 30.23 ± 9.87 years) agreed to participate in the study. Among them, 99 patients were referred for dental treatment under GA due to dental anxiety, while the other three patients were referred due to systemic disorders. Out of the 102 patients, 71 included the self-completed IDAF-4C+ questionnaire. The mean IDAF score from Module 1 was 33.21 ± 7.2, with 67 patients scoring over 20, and 7.1 ± 1.5 from Module 2, with 25 patients scoring over 7.5. All of the included patients participated in the preoperative evaluation (T0; Group 1). Among them, 15 avoided the GA intervention and 87 were treated under GA. Out of all participants, 36 participated in the one-month postoperative evaluation (Group 2), and 15 were evaluated six months postoperatively (Group 3). The flow diagram of participation in the study is shown in Figure 1.

Sociodemographic data and mean values for dental status, GOHAI score, and mastication criteria recorded at T0 for the three groups are collected in Table 3. Patients failing the GA appointment for dental treatment had similar GOHAI scores to those being treated.

At T0, patients had a high DMFT score with a significant D component, a low number of PFU, and a poor oral health-related quality of life. Moreover, they were unable to adapt their chewing strategy to the food texture, as their chewing frequency decreased when chewing hard gelatin compared to soft gelatin. The mean D50 value of raw carrot was higher than the MNI limit of 4 mm, and it was negatively correlated with the number of PFU (r = −0.2941, *p* = 0.0054).

Independent intergroup comparisons of the preoperative data showed that the group of 66 patients who were evaluated at T0 and underwent GA but missed the T1 one-month postoperative evaluation were similar in terms of age, dental status, and chewing parameters to the group of 36 patients who participated in the T1 evaluation. The group of 36 patients evaluated at T1 was similar for all preoperative data to the group of 15 patients who completed all evaluations up to T2.

Data comparisons between T0 and T1 (Table 4) demonstrated that: (i) the OHRQoL improved in every GOHAI domain, with the ADD-GOHAI score increasing from “low” to “moderate” level; (ii) the chewing frequency of raw carrot increased; (iii) the number of PFU did not change before and after GA, despite an average number of three extractions per AG; (iv) the number of chewing cycles and chewing time of hard gelatin increased, characterizing improved chewing efficiency; (v) chewing frequency did not change between hard and soft gelatin, indicating good adaptive abilities; and (vi) raw carrot bolus granulometry decreased slightly, but the difference was not statistically significant.

## 4. Discussion

This study evaluated the effects of a comprehensive and conservative approach to dental treatment under GA on oral functioning. As expected, oral health-related quality of life improved in three GOHAI domains. Moreover, mastication function did improve unexpectedly, despite the requirement of teeth extractions and the application of adapted restorative procedures. This study brought to light several points of discussion.

Measuring the impact of dental treatment under GA would require a detailed analysis of the dental procedures applied during GA. Obviously, extractions of teeth with deep carious lesions would lead to pain being eliminated, which in turn, strongly improves oral health-related quality of life for the patients and their families. However, eliminating pain is not the only goal of dental treatment, and all patients should have the same opportunity to access the same quality of treatment, regardless of the conditions for accessing this treatment. The interpretation of the expression “dental treatment under GA” is often confusing for the general population, dental extractions being considered as dental care. Few studies were designed to measure the benefit for the patients to treat their teeth rather than to extract them under GA. Positive changes in the oral health-related quality of life after dental treatments under general anesthesia were previously reported in adult patients with learning disabilities with the use of self- and proxy-questionnaires. It was detailed that the increase in COHIP-14 or OHIP-14 scores after GA was higher when a low number of teeth were extracted, and lower when teeth were endodontically treated [32,33]. The present study concurs with these findings. However, other authors suggested that subjective and objective evaluations of mastication were not correlated [34,35]. The masticatory performance was assessed by the determination of the individual capacity of fragmenting test materials based on silicone, plaster, and tooth paste, and by the heterogeneity of a color-changeable gum chewed for one minute. Chewing gum and test materials are not brain-managed in the same way than real foods [36]. Moreover the subjects were 32 healthy and fully dentate adolescents. One can hypothesize that the correlations between subjective and objective measurements of mastication depend on methodological conditions. The present study concurs with previous ones, validating improvements of mastication based on objective physiological criteria in correlation with improvements of oral health quality of life scores. Preserving at least four pairs of functioning posterior teeth is important to chewing activity. Indeed, during mastication, teeth reduce the food to an ingestible bolus and assist in regulating the chewing activity according to food texture via the mechanical receptors localized in the periapical structures [36,37].

The adapted conservative procedures applied in this comprehensive approach, such as pulpotomies and restorations with preformed stainless steel crowns (SSCs) on permanent mature teeth, could be criticized, as they are not referred among conventional definitive dental treatments. Over the last decade, in relation to a better understanding of pulp biology, pulpotomy has been reinvestigated as a definitive treatment for mature permanent teeth and indications for pulpotomy in mature permanent teeth presenting reversible or irreversible pulpitis are now widely debated [38]. In past years, convergent results from the literature suggested that permanent vital teeth with pulpitis may be treated using full pulpotomy. Several studies have shown similar success rates for full pulpotomy compared to root canal treatment (RCT) [39]. It might be expected that full pulpotomy will become the endodontic treatment of choice for affected teeth with a vital pulp, instead of RCT. This may be particularly true when treatment is undertaken in special conditions limiting the operatory time and subsequent intervention, as under GA. A recent study reported that among 263 teeth treated with full pulpotomy under GA and evaluated after a median follow-up period of 24 months, 89% were effective, 7.6% were of uncertain outcome, and 3.4% were ineffective. Neither the etiology of lesion, the tooth maturity, nor the endodontic difficulty influenced the effectiveness rate.

Preformed crowns provide full-coverage coronal restorations that reduce the risk of recurrent caries in extensively decayed teeth. For permanent teeth, SSCs were considered to be interim restorations [40]. Two main arguments are opposed to considering SSCs as definitive restoration. Firstly, SSCs are prefabricated and do not reproduce the features’ tooth specific to each individual. Consequently, one can suspect that the poor adaptation of the cervical margin, and of the proximal and occlusal contacts, could cause periodontal diseases. Secondly, the stainless steel is thin and the occlusal surface could be perforated over time. Few studies reported data on the evaluation of SSC restorations on permanent teeth. The long-term outcome of 766 preformed crowns was evaluated radiographically in 271 adult patients with special needs treated under GA [41]. It was shown that the 10-year survival rate for SSCs was 79.2% and the mean alveolar bone loss on the sites recorded was less than 2 mm. Compared to conventional crowns, the occlusal cusp volumes of SSCs are standardized and interlocking with antagonist teeth is poor, which gives rise to questions on the impact of mastication performance. The present study showed that chewing efficiency improved, with an increase in chewing frequency and a decrease in the carrot bolus granulometry. This suggests that SSC restorations compensate for extractions by renewing inter-arch contacts. The decrease in carrot bolus granulometry observed after treatments also suggested that nutrients’ bio-accessibility could also be improved [42]. Follow-up studies should evaluate whether nutritional changes could occur when improving mastication in treated patients.

Preserving teeth using alternative procedures allows the practitioner to offer patients the opportunity to achieve root canal treatment in case of failed pulpotomy or to replace preformed crowns with better quality prosthetics at a later stage. This does not represent a loss of opportunity in terms of public health as patients retain all the benefits of having kept their teeth and will be able to access conventional dentistry in the dental chair upon successfully managing their dental anxiety and/or having sufficient financial resources. However, this study also reported that the majority of patients treated under GA did not attend their postoperative appointments, and preformed crowns were rarely replaced with conventional fixed prosthetic crowns. Additional studies are needed to determine the tooth survival rate and the degree of periodontal morbidity induced by the preformed crowns.

Discussions should not lead to criticism over widely evaluated and successful academic conventional dental treatments. The notion that the improvement of oral health derives from developing dental technologies is supported worldwide. However, that could only be verified in patients who are able to access such treatments. Considering the benefits reported here for uncooperative patients treated under GA, it could be suggested that this approach to treatment, developed to maintain teeth on the arches, could be recommended in settings other than the operating room. Indeed, other populations, such as low-income patients, patients with neuromotor, mental, or sensorial disabilities, homeless people, or inmates could also benefit from such an approach.

The main study limitations are related to the short follow-up period, and to the high number of “lost to follow-up” patients, implying a five-year study to obtain the required number of subjects evaluated six months postoperatively. This study received ethical approval as an observational study of routine dental treatment. Under these conditions, it was not possible to recall the patients who missed their regular appointments. After inclusion, 15 patients did not attend the GA appointment for dental treatment, 51 were lacking the postoperative examination that would be due one month postoperatively, and finally, 21 patients were lost before the six-month postoperative follow-up. The return rate, calculated from the pool of included patients, was very low (15%). This could be analyzed by considering the patients’ behavior faced with dental treatment. It was previously described that patients with dental anxiety missed their appointments [43,44,45,46].

Excluding patients over 65 years old and those with disabilities could be discussed. Three factors have a major impact on masticatory function in elderly persons: the number of natural antagonist teeth, the quantity or/and quality of saliva, and the impairment of the motor apparatus. Each of these three factors is largely correlated with ageing and, from a pragmatic point of view, need to be included when considering mastication in the elderly [47]. To avoid confounding factors related to ageing, elderlies were excluded based on the 65 years old cut-off value considered by the World Health Organization to discriminate old patients. Excluding patients with disabilities is another point of discussion. The same conservative approach was performed for all 367 patients with disabilities treated under GA during the study period, and it would be of interest to reinforce ethical principles of social inclusion by including these patients. However, the study was based on psychometric and physiological data that could not be collected with the same methodologies than those for included patients. It should be emphasized that this translational study encompassed all patients treated under GA.

Patients accessing dental care under general anesthesia have very diverse biopsychosocial situations. Among them, patients with dental anxiety have special needs that require both a dental professional and a psychological therapist. The dental fear cycle was well-studied by David Locker, and behavioral and cognitive therapies (BCT) are indicated to treat dental anxiety [48]. During BCT, patients need to be exposed to dental care progressively, in a positive situation without pain exposition nor negative judgment. He/she had to understand that he/she could have the control of the situation. Treating these patients under general anesthesia avoids exposition to dental care and reinforcing anxiety. However, BCT programs require time and financial means that patients with dental anxiety do not have in most cases. Indeed, and particularly for patients with acute dental disease, GA represents the only way to access dental care. It is then up to the dental professional to refer the patient to a therapist to treat his/her dental anxiety after oral rehabilitation under GA. The benefits brought to the patient in terms of quality of life and chewing ability can be influential factors to bring back the patient in a dental chair care setting.

## 5. Conclusions

This study reports the improvement of oral health quality of life and mastication function through oral rehabilitation for patients treated under general anesthesia. Oral health-related quality of life improved in three GOHAI domains. Moreover, mastication function did improve unexpectedly, despite teeth extractions and the application of adapted restorative procedures, with both factors being challenging for occlusion rehabilitation. Further studies are now needed to evaluate whether oral health improvements would induce in turn changes on nutritional status of these patients.

The results of this study are not restricted to the conditions of patients with dental anxiety, as they encompass the situation of any patients being treating under GA. Dental services could be organized to generalize conservative dental care under general anesthesia for uncooperative patients. To achieve this goal, dental professionals should be taught to the comprehensive strategies that limits extractions to the profit of conservative treatments. Qualitative studies are now needed to describe the patients’ views of such an approach.

## Figures and Tables

**Figure 1 ijerph-17-07336-f001:**
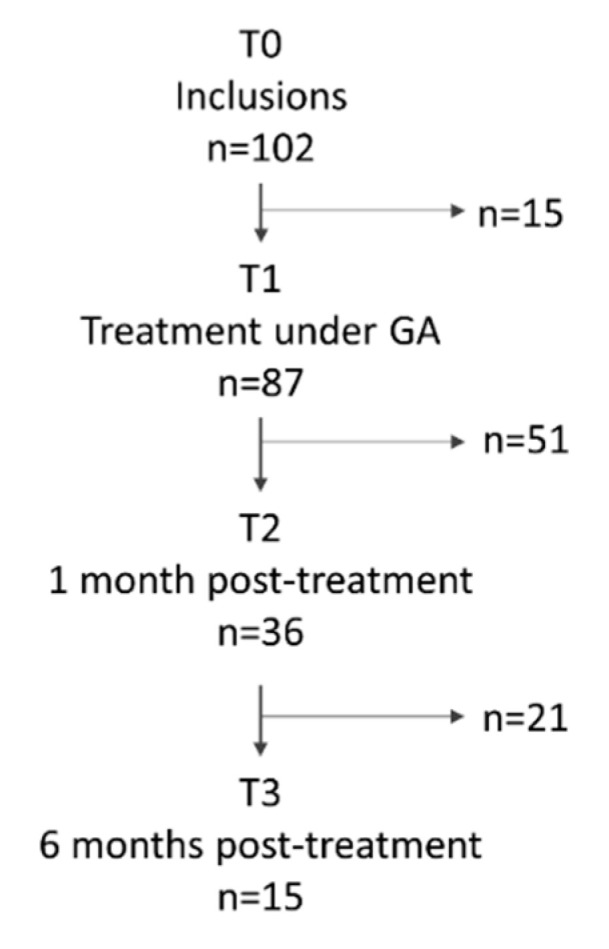
Flow chart of participation.

**Table 1 ijerph-17-07336-t001:** Description of procedures for dental treatment under general anesthesia.

General	
	Operator: dental surgeon senior
Operators	Assistant: dental surgeon junior or resident
	Non-sterile operating aid: operating room nurse
Suction	Surgical suction
**Endodontic Treatments**	
Isolation	Rubber dam with vinyl foils
Removal of carious dentin	Use of tungsten carbide bur
Preparation of access cavity	Use of Endo-Z^®^ bur
Root canal treatment	
Indications	Visible apical lesion on the radiogram (PAI score >2 for at least one root) or pre-operative exposure of endocanal to saliva or lack of bleeding after per-operative pulp opening
Per operative radiography	Argentic and digital radiography with or without sensor holder
Work length determination	Per operative radiography or electronic measurement
Irrigation and disinfection	2.5% Sodium hypochlorite solution;
Root canal preparation	Step down with manual files (Gates 4-3-2, lime K 15-20-25-30-35) or NiTi files (Protaper^®^, RevoS^®^, or Wave one^®^)
Drying	Sterile paper points
Root canal obturation	One gutta percha master point with zinc oxide eugenol sealer and thermomechanical compaction
Coronal restoration	Composite, GIC, bonded amalgams with GIC, Stainless Steel Crown depending on teeth and carious lesion extension
Full pulpotomy	
Indications	Absence of apical lesion on the radiogram (PAI score <2 for all roots) Bleeding after pulp opening
Coronal one third preparation	Gates Glidden burs (4-3)
Irrigation and disinfection	2.5% Sodium hypochlorite solution
Hemostasis	Compression with sterile cotton pellet; Calcium hydroxide in cases of abundant bleeding
Pulp capping	Reinforced zinc oxide eugenol material (IRM^®^)
Pulp chamber obturation	High viscosity GIC (Fuji IX^®^)
Coronal restoration	Stainless Steel Crown for molars; bonded amalgams with GIC or composite based on GIC for premolars
**Conservative treatments**	
Incisor or canine	Composite resin
Bicuspid	With one or two marginal crest: GIC bonded Amalgam or Composite resin
	Without marginal crest: Stainless Steel Crown, GIC bonded Amalgam or Composite resin
	After endodontic treatment: Stainless Steel Crown
Molar	With one or two marginal crest: GIC bonded Amalgam
	Without marginal crest or after pulpotomy or endodontic treatment: Stainless Steel Crown
Cervical cavity	GIC bonded Amalgam or Composite resin
Extractions	
Indications	Periodontal disease with mobility Periodontal involvement of the furcation Extensive carious lesion with root and/or furcation involvement

GIC: glass ionomer cement; PAI: Periapical index [25]; NiTi: Nickel-Titan; IRM: intermediate restorative material.

**Table 2 ijerph-17-07336-t002:** Criteria for inclusion and exclusion.

Inclusion criteria	Patients aged from 18 to 65 years old. Adult patient referred for dental care under GA due to lack of cooperation for dental care un the dental chair. Patients with at least one decayed tooth at stage 3 or 4 on the Ekstrand Classification of caries progression on each quadrant. Patients agreeing to the use of the data collected during their dental treatment.
Exclusion criteria	Patients being less than 18 years old and those over 65 years old. Patients being referred for full clearance under GA. Patients with learning disabilities. Patients who did not speak French. Patients without medical insurance. Patients with neuromotor and/or cognitive disabilities affecting their capacities to chew and spit. Patients with behavior problems possibly affecting their participation to the study data collection. Patients whose treatment required two or more sessions of general anesthesia. Patients disagreeing for the use of the data collected during their dental treatment.

GA: General Anaesthesia.

**Table 3 ijerph-17-07336-t003:** Comparison of patients being treated at T0 but not evaluated after 1 month (G1–G2), those being treated and evaluated one month after GA (G2), those being evaluated at T2 but not at T3 (G2–G3) and those being evaluated 6 months after treatment (G3) in terms of age, sex, carious index, quality of life, and parameters of mastication before care under GA.

	G1–G2 (*n* = 66)	G2 (*n* = 36)	Student T-Test	G2–G3 (*n* = 21)	G3 (*n* = 15)	Student T-test
Mean age (years)	31.24 ± 10.44	28.29 ± 8.64	NS	28.37 ± 8.08	28.17 ± 9.66	NS
Male-to-female ratio	36/30	7/29	*p* < 0.001	4/17	3/12	NS
Carious index (DMFT)						
Number of decayed teeth (D)	14.19 ± 5.93	11.81 ± 5.47	NS	11.67 ± 5.50	12.00 ± 5.61	NS
Number of missing teeth (M)	3.49 ± 3.35	2.97 ± 2.43	NS	2.81 ± 2.11	3.20 ± 2.88	NS
Number of filling teeth (F)	2.07 ± 2.68	3.00 ± 4.11	NS	3.52 ± 4.72	2.27 ± 3.08	NS
DMFT (teeth)	18.77 ± 7.51	17.78 ± 5.76	NS	18.00 ± 5.99	17.47 ± 5.60	NS
Number of posterior functional unit (PFU)	4.63 ± 2.22	5.83 ± 2.29	*p* = 0.012	5.81 ± 2.23	5.87 ± 2.47	NS
Quality of life (GOHAI score)						
Functional field (max: 20)	14.28 ± 3.43	14.38 ± 3.76	NS	14.52 ± 4.25	14.47 ± 3.16	NS
Psychosocial field (max: 25)	13.39 ± 4.55	15.76 ± 5.34	NS	16.62 ± 5.201	14.60 ± 5.12	NS
Pain or discomfort field (max: 15)	7.19 ± 2.69	8.67 ± 2.95	*p* = 0.013	8.81 ± 2.66	8.27 ± 3.35	NS
Total (max: 60)	34.86 ± 7.72	38.82 ± 10.36	*p* = 0.022	39.95 ± 10.34	37.33 ± 10.11	NS
Chewing parameters (raw carrot at swallowing time)						
Number of chewing cycles	48.23 ± 19.66	50.68 ± 47.21	NS	40.17 ± 19.80	62.58 ± 64.81	NS
Chewing time (s)	32.65 ± 14.64	35.08 ± 34.27	NS	27.55 ± 15.67	43.62 ± 46.61	NS
Chewing frequency (cycles/s)	1.50 ± 0.21	1.48 ± 0.23	NS	1.49 ± 0.27	1.47 ± 0.18	NS
Bolus granulometry D50 (µm)	8561 ± 2800	8294 ± 3045	NS	8957 ± 3221	7410 ± 2640	NS
Chewing adaptation to food hardness parameters						
Soft test food						
Number of chewing cycles	22.59 ± 14.01	22.23 ± 14.75	NS	20.58 ± 12.70	28.73 ± 21.86	NS
Chewing time (s)	19.08 ± 13.95	18.17 ± 12.56	NS	16.94 ± 10.73	23.33 ± 18.74	NS
Chewing frequency (cycles/s)	1.24 ± 0.24	1.25 ± 0.23	NS	1.23 ± 0.27	1.26 ± 0.16	NS
Hard test food						
Number of chewing cycles	29.09 ± 13.06	30.63 ± 20.32	NS	29.33 ± 18.75	35.63 ± 23.39	NS
Chewing time (s)	24.29 ± 10.58	25.57 ± 17.71	NS	24.64 ± 16.43	29.17 ± 19.90	NS
Chewing frequency (cycles/s)	1.23 ± 0.21	1.22 ± 0.23	NS	1.21 ± 0.23	1.24 ± 0.21	NS
Comparison of chewing frequency for hard/soft gelatins	NS	NS		NS	NS	

DMFT index: sum of the number of decayed, missed and filled teeth; NS: non significant.

**Table 4 ijerph-17-07336-t004:** Comparisons of study criteria values for the group of patients evaluated before treatment under GA (T0) and one month after treatment (T1) (repeated measures ANOVA with post hoc Bonferroni correction).

GROUP 2	T0	T1	Repeated Measured Process		
			F	PES	Significant
Carious index (DMFT)					
Number of patients (n)	36	36			
Number of decayed teeth (D)	11.81 ± 5.47	0.25 ± 0.91	181.52	0.84	*p* < 0.001
Number of missing teeth (M)	2.97 ± 2.43	5.86 ± 3.27	39.90	0.53	*p* < 0.001
Number of filling teeth (F)	3.00 ± 4.11	11.75 ± 5.07	124.17	0.79	*p* < 0.001
DMF (teeth)	17.78 ± 5.76	17.86 ± 5.64	0.81	0.02	NS
Number of Posterior Functional unit (PFU)	5.83 ± 2.29	5.78 ± 2.27	0.70	0.004	NS
Quality of life (GOHAI score)					
Number of patients (n)	36	36			
Functional field (max: 20)	14.38 ± 3.76	17.62 ± 2.76	28.92	0.47	*p* < 0.001
Psychosocial field (max: 25)	15.76 ± 5.34	22.24 ± 3.67	72.22	0.69	*p* < 0.001
Pain or discomfort field (max: 15)	8.67 ± 2.95	12.24 ± 2.23	47.31	0.59	*p* < 0.001
Total (max: 60)	38.82 ± 10.36	52.09 ± 7.48	93.80	0.74	*p* < 0.001
Chewing parameters (raw carrot at swallowing time)					
Number of patients (n)	35 (1 refusal)	35 (1 refusal)	35		
Number of chewing cycles	50.68 ± 47.21	51.09 ± 54.54	2.04	0.07	NS
Chewing time (s)	35.08 ± 34.27	34.38 ± 42.19	0.58	0.02	NS
Chewing frequency (cycles/s)	1.48 ± 0.23	1.57 ± 0.25	10.17	0.26	*p* = 0.003
Bolus granulometry—D50 (µm)	8294 ± 3045	7727 ± 2683	2.13	0.06	NS
Chewing adaptation to food hardness parameters					
Number of patients (n)	33 (3 refusal)	33 (3 refusal)	31		
Soft test food					
Number of chewing cycles	24.29 ± 17.65	24.39 ± 22.75	1.31	0.04	NS
Chewing time (s)	19.85 ± 15.00	19.08 ± 18.62	0.50	0.02	NS
Chewing frequency (cycles/s)	1.25 ± 0.22	1.31 ± 0.23	1.88	0.06	NS
Hard test food					
Number of chewing cycles	32.20 ± 20.89	42.00 ± 37.23	9.74	0.25	*p* = 0.004
Chewing time (s)	26.70 ± 17.94	33.44 ± 31.57	7.94	0.21	*p* = 0.008
Chewing frequency (cycles/s)	1.22 ± 0.22	1.28 ± 0.23	4.02	0.12	NS
Comparison of chewing frequency for hard/soft gelatins	NS F = 4,10 PES = 0.04	NS F = 2.48 PES = 0.07			
Operating Time (min)	115.5 ± 42.1				
Treatments performed during: Tooth extraction					
Incisor or canine	0.22 ± 0.48				
Molar or premolar	2.72 ± 3.05				
Impacted or retained tooth	0.12 ± 0.45				
Total	3.52 ± 3.19				
Conservative treatments					
Composite restoration	2.61 ± 3.72				
Bonded amalgam restoration	3.69 ± 2.66				
Stainless steel crown	0.39 ± 0.69				
Glass ionomer cement restoration	0.33 ± 1.53				
Stainless steel crown and endodontic treatment	0.65 ± 0.75				
Bonded amalgam restoration and endodontic treatment	0.12 ± 0.40				
Composite restoration and endodontic treatment	0.81 ± 1.37				
Composite restoration with fiber post and endodontic treatment	0.02 ± 0.15				
Composite and pulpotomy	0.12 ± 0.45				
Bonded amalgam and pulpotomy	0.12 ± 0.40				
Stainless steel crown and pulpotomy	0.67 ± 0.98				
Ultrasonic scaling	0.63 ± 0.49				

Data comparisons between T1 and T2 are collected in Table 5. No significant changes appeared in the long-term postoperative period, although comparisons of bolus granulometry between T0 and T2 showed a statistically significant decrease, while remaining above the masticatory normal indicator (MNI) threshold. This decrease confirmed improvement of the mastication efficiency.

**Table 5 ijerph-17-07336-t005:** Comparison of study criteria describing patients in Group 3 at different times of evaluation (repeated measures ANOVA with post hoc Bonferroni correction).

GROUP 3	T0	T1	T2	Repeated Measured Process
F	PES	*p*
**Carious index (DMFT)**		
Number of patients (n)	15	15	15		
Number of decayed teeth (D)	12.00 ± 5.61	0 ± 0	0 ± 0	68.72	0.83	*p* < 0.001
Number of missing teeth (M)	3.20 ± 2.88	5.60 ± 3.78	5.60 ± 3.78	7.68	0.35	*p* = 0.002
Number of filling teeth (F)	2.27 ± 3.08	11.93 ± 4.43	11.93 ± 4.43	48.90	0.78	*p* < 0.001
DMF (teeth)	17.47 ± 5.60	17.53 ± 5.33	17.53 ± 5.33	0.13	0.01	NS
**Number of posterior functional unit (PFU)**	5.87 ± 2.47	5.60 ± 2.32	6.21 ± 2.01	1.11	0.07	NS
**Quality of life (GOHAI score)**
Number of patients (n)	15	15	15		
Functional field (max: 20)	14.47 ± 3.16	17.46 ± 2.47	18.88 ± 2.10	10.56	0.43	*p* < 0.001
Psychosocial field (max: 25)	14.60 ± 5.12	21.85 ± 4.18	21.63 ± 5.07	18.96	0.58	*p* < 0.001
Pain or discomfort field (max: 15)	8.27 ± 3.35	11.69 ± 2.50	13.00 ± 2.45	9.98	0.42	*p* = 0.001
Total (max: 60)	37.33 ± 10.11	51.00 ± 7.44	53.50 ± 9.17	19.70	0.59	*p* < 0.001
**Chewing parameters (raw carrot at swallowing time)**
Number of patients (n)	15	15	15		
Number of chewing cycles	62.58 ± 64.81	63.85 ± 75.34	57.93 ± 38.64	0.61	0.05	NS
Chewing time (s)	43.62 ± 46.61	43.15 ± 58.41	39.67 ± 35.78	1.1	0.17	NS
Chewing frequency (cycles/s)	1.47 ± 0.18	1.59 ± 0.21	1.59 ± 0.25	5.66	0.32	*p* = 0.01
Bolus granulometry—D50 (µm)	7410 ± 2634	6587 ± 1779	6276 ± 2126	4.01	0.22	NS
**Chewing adaptation to food hardness parameters**
Number of patients (n)	15	13 (2 refusals)	15		
**Soft test food**						
Number of chewing cycles	28.73 ± 21.86	28.81 ± 29.94	36.77 ± 54.53	0.60	0.10	NS
Chewing time (s)	23.33 ± 18.74	22.27 ± 24.36	30.10 ± 50.02	1.06	0.08	NS
Chewing frequency (cycles/s)	1.26 ± 0.16	1.34 ± 0.21	1.32 ± 0.20	2.61	0.18	NS
**Hard test food**						
Number of chewing cycles	35.63 ± 23.99	51.58 ± 47.30	45.90 ± 24.60	4.21	0.26	NS
Chewing time (s)	29.17 ± 19.90	39.85 ± 37.91	36.53 ± 23.74	4.14	0.26	NS
Chewing frequency (cycles/s)	1.24 ± 0.21	1.33 ± 0.21	1.31 ± 0.24	3.01	0.20	NS
**Comparison of chewing frequency for hard/soft gelatins**	NS	NS	NS		
**Operating Time (min)**	123.3 ± 36.9	
**Treatments performed during the procedure** **Tooth extraction**		
Incisor or canine	0.2 ± 0.41	
Molar or bicuspid	2.3 ± 3.62	
Impacted or retained tooth	0.13 ± 0.52	
Total	2.67 ± 3.77	
**Conservative treatments**		
Composite restoration	3.87 ± 5.11	
Bonded amalgam restoration	3.47 ± 2.33	
Stainless steel crown	0.20 ± 0.56	
Glass ionomer cement restoration	0.07 ± 0.26	
Stainless steel crown and endodontic treatment	0.47 ± 0.52	
Bonded amalgam restoration and endodontic treatment	0.20 ± 0.41	
Composite restoration and endodontic treatment	0.60 ± 0.74	
Composite restoration with fiber post and endodontic treatment	0	
Composite and pulpotomy	0	
Bonded amalgam and pulpotomy	0.07 ± 0.26	
Stainless steel crown and pulpotomy	1.00 ± 1.13	
**Ultrasonic scaling**	0.53 ± 0.52

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
