# Peer review of "A Comprehensive Approach Limiting Extractions under General Anesthesia Could Improve Oral Health"

_ijerph, 2020, doi:10.3390/ijerph17197336_

Round 1

Reviewer 1 Report

Strong and sound study, the scientific literature in need for it. Well done. 

Abstract: 

Please do the changes highlighted in the reviewed manuscript (lines 18 & 19) 

And please add the age range cause its really varies. 

Intro: 

Lines 60 & 61: The treatments that are provided under GA are the same treatments that are provided in the clinic, but in an advantage that it can be done easily and without interruption by the patients. So it always done with higher quality under GA. 

Check the highlighted lines and comments on the reviewed manuscripts. 

Please mention clearly the exclusion and inclusion criteria.

Methods:

Why it takes you from 2015 till now 20202 to publish this study ???

Please mention clearly the exclusion and inclusion criteria.

Line 108: Dental status was characterised using the DMFT index 2003!! why the authors did not follow the recent published DMT index. ?

Discussion:

The first paragraph: is pointing that this is the first study that "This is the first study evaluating the effects of a comprehensive and conservative approach to dental treatment under GA on oral functioning"

So i prefer to keep it in a more moderate pace, specially that I'm sure that the authors did not review all the studies that written in other languages.... 

Discussion should always be long and discuss the important finings of the cohort;

so i wish if the authors can elaborate more about the point that mentioned in in the paragraph between lines 273 and 276.

Conclusion:

Authors should answer their hypothesis clearly in their conclusion, so please answer the hypothesis mentioned in line 186.

References:

Are too much we can make it less, but its ok for me please refer to the editors comment about that.

Author Response

Abstract:

Please do the changes highlighted in the reviewed manuscript (lines 18 & 19)

Answer:

Not all included patients were nasally intubated. This point was discussed in the intro discussion section, lines 61-74

And please add the age range cause its really varies “between 18-65 y old”

Answer: the range was added.

102 adult patients (mean age ± SD: 32.2±99 years; range: 18-57.5)…

Intro:

Lines 60 & 61: The treatments that are provided under GA are the same treatments that are provided in the clinic, but in an advantage that it can be done easily and without interruption by the patients. So it always done with higher quality under GA.

Check the highlighted lines and comments on the reviewed manuscripts.

Answer:

The sentence was deleted.

These concerns were added in the Introduction as follows (lines 61-74)

It was reported that GA facilitates achieving dental treatment in uncooperative children [18]. However, the full mouth rehabilitation in adults is more difficult for several reasons. First, the dental treatments on permanent teeth have to survive for the entire life span of the individual, while the expected survival time on deciduous teeth is less than 10 years. Second, in most of countries, when outpatients are treated in one-day surgery, the duration of the intervention is limited to two hours. However, treating 32 teeth required a longer time than for treating 20 teeth. Third, depending on the countries, the anaesthetists’ team, the equipment and the clinical cases, patients could be intubated either nasally or orally. Recording the patient bite to adjust the occlusal restoration is always difficult under GA, even when patient are intubated orally as the tube and the packing put the tongue and the jaw forward. In case of oral intubation, it is necessary to move the tube on one side in a retromolar position in order to guide biting. Such manipulations cost additional time. For all these reasons, it is legitimate to doubt the outcome of occlusal restorative procedures performed under GA. Measuring changes in the chewing ability in patients treated under GA would answer these questions.

  1. Eidelman E, Faibis S, Peretz B. A comparison of restorations for children with early childhood caries treated under general anesthesia or conscious sedation. Pediatr Dent, 2000, 22(1), 33-7.
  2. Jockusch J, Hopfenmüller W, Ettinger R, Nitschke I. Outpatient, dental care of adult vulnerable patients under general anaesthesia-a retrospective evaluation of need for treatment and dental follow-up care. Clin Oral Investig. 2020 Sep 15. doi: 10.1007/s00784-020-03564-2. Epub ahead of print.
  3. Mallineni SK, Yung Yiu CK, "A Retrospective Review of Outcomes of Dental Treatment Performed for Special Needs Patients under General Anaesthesia: 2-Year Follow-Up". ScientificWorldJournal 2014;2014:748353. https://doi.org/10.1155/2014/748353

Please mention clearly the exclusion and inclusion criteria.

Answer:

We included inclusion and exclusion criteria in an additional table in the method section.

Methods:

Why it takes you from 2015 till now 2020 to publish this study ???

Answer:

We agreed with Reviewer#1, the time between the end of the study and the present submission is very long. The writing of the manuscript was carried out over a period 18 months inside which the first author had to defend his thesis. Since then, we have submitted this article to Journal of Dental Research, Clinical Oral Investigation, Journal of Oral Rehabilitation and Journal of Dental Research Clinical and Translational Research. Depending on the journal, the decision and/or the review processing took one week to 8 months.

Please mention clearly the exclusion and inclusion criteria.

 Answer:

The following table was inserted table 2 in lines (139-141)

Inclusion criteria

· Patients aged from 18 to 65 years old

· Adult patient referred for dental care under GA due to lack of cooperation for dental care un the dental chair

· Patients with at least one decayed tooth at stage 3 or 4 on the Ekstrand Classification of caries progression on each quadrant.

· Patients agreeing to the use of the data collected during their dental treatment

Exclusion criteria

·  Patients being less than 18 years old and those over 65 years old

·  Patients being referred for full clearance under GA

·  Patients with learning disabilities

·  Patients who did not speak French

·  Patients without medical insurance

·  Patients with neuromotor and/or cognitive disabilities affecting their capacities to chew and spilt

·  Patients with behaviour problems possibly affecting their participation to the study data collection.

·  Patients whose treatment required two or more sessions of general anaesthesia.

·  Patients disagreeing for the use of the data collected during their dental treatment

Line 108: Dental status was characterised using the DMFT index 2003!! why the authors did not follow the recent published DMT index. ?

Answer:

We began the study before the updated DMT was published in 2013 (WHO Fifth edition). Moreover, DMT is more detailed than DMFT and its values can not be deducted from DMFT.

Discussion:

The first paragraph: is pointing that this is the first study that "This is the first study evaluating the effects of a comprehensive and conservative approach to dental treatment under GA on oral functioning"

So i prefer to keep it in a more moderate pace, specially that I'm sure that the authors did not review all the studies that written in other languages....

Answer:

We agree with reviewer#1, we referred to our knowledge, which is necessarily incomplete. We changed the sentence as follows;

This is the first study evaluated the effects of a comprehensive and conservative approach to dental treatment under GA on oral functioning.

Discussion should always be long and discuss the important findings of the cohort;

so i wish if the authors can elaborate more about the point that mentioned in in the paragraph between lines 273 and 276.

Answer : we include the following text from line 306.

The present study concurs with these findings. However, other authors suggested that subjective and objective evaluations of mastication were not correlated [34,35]. The masticatory performance was assessed by the determination of the individual capacity of fragmenting test materials based on silicone, plaster and tooth paste, and by the heterogeneity of a colour-changeable gum chewed during one minute. Chewing gum and test materials are not brain managed in the same way than real foods [36]. Moreover the subjects were 32 healthy and fully dentate adolescents. One can hypothesize that the correlations between subjective and objective measurements of mastication depend on methodological conditions.

  1. Bourdiol P, Hennequin M, Peyron MA, Woda A. Masticatory Adaptation to Occlusal Changes. Front Physiol. 2020 3, 11, 263. doi: 10.3389/fphys.2020.00263

Conclusion:

Authors should answer their hypothesis clearly in their conclusion, so please answer the hypothesis mentioned in line 186.

Answer (we supposed the line was 168).

We inserted the following sentences at the beginning of the conclusion section:

 This study reported the improvement of oral health quality of life and mastication function through oral rehabilitation for patients treated under general anaesthesia. Oral health-related quality of life improved in three GOHAI domains. Moreover, mastication function did unexpectedly improve, despite teeth extractions and the application of adapted restorative procedures conditions of dental treatment under GA, both factors being challenging for occlusion rehabilitation. Further studies are now needed to evaluate whether oral health improvements would, in turn, induced changes on the nutritional status of these patients.  

The results of this study are not restricted to the conditions of patients with dental anxiety as they encompass the situation of any patients being treating under GA.

References:

Are too much we can make it less, but its ok for me please refer to the editors comment about that.

Reviewer 2 Report

The authors report in their study on the improvement of OHRQoL through oral rehabilitation of uncooperative patients under general anesthesia.
To my knowledge, this is the first study to investigate the positive effects of rehabilitation of these particular patients. As a clinician, I can say that these patients regularly attend clinics and often, regrettably, extensive tooth extractions occur although conservative measures are still possible.
All the more I am pleased that the positive effect of tooth preserving measures on OHRQoL could be demonstrated here by a practicable concept. A few minor spelling errors (e.g. line 78; 80) should be corrected. Line 63 "Materials and Methods" should also be listed as a separate heading.

Author Response

The authors report in their study on the improvement of OHRQoL through oral rehabilitation of uncooperative patients under general anesthesia.

To my knowledge, this is the first study to investigate the positive effects of rehabilitation of these particular patients. As a clinician, I can say that these patients regularly attend clinics and often, regrettably, extensive tooth extractions occur although conservative measures are still possible.

All the more I am pleased that the positive effect of tooth preserving measures on OHRQoL could be demonstrated here by a practicable concept. A few minor spelling errors (e.g. line 78; 80) should be corrected.

Line 63 "Materials and Methods" should also be listed as a separate heading:

Answer:

Done

Reviewer 3 Report

Peer review of manuscript:                                                                September 2020

Manuscript ID: ijerph-928276

Title: A comprehensive approach limiting extractions under general anaesthesia could improve oral health

Journal: International Journal of Environmental Research and Public Health

Specific comments:

Line 86 For patients requiring multiple dental treatments, two consecutive interventions were organized at a four to six months interval.

How did you deal with more than one intervention when you did the calculations? The chewing ability probably change over time after the first and second treatment. I can´t find any comments under the result section, missing?

Line 90 Patients who demonstrated high levels of anxiety received 5 ml of midazolam orally, before going into the operating room

Concentration of midazolam is lacking!

Line 101 The next day, a nurse phoned them to check for any adverse reactions to the intervention.

I can not see any comments regarding this in the results, are there something to add?

Line 107 The cohort was populated by uncooperative adult patients (between 18 and 65 years old) referred by general dentists or physicians to the Special Care Unit of the Clermont-Ferrand University Hospital after failure to be treated in the dental chair, from 01/01/2011 to 31/12/2015.

Why exclusion of patients over 65 year if they are healthy? You have 102 patients in the results (Line 195) who were excluded because of age but they got treatment under GA so the exclusion criteria were not the risk within the GA. I suggest that you follow it up in the discussion.

Line 285 Over the last decade, in relation to a better understanding of pulp biology, pulpotomy has been reinvestigated as a definitive treatment for mature permanent teeth and indications for pulpotomy in mature permanent teeth presenting reversible or irreversible pulpitis are now widely debated [35].

Consider a discussion about the questions below:

Did you have any postoperative complications? Non are reported

Comments about pulpotomy prognosis in adults compared to children?

Wouldn´t be better to perform more of root canal treatment directly instead of pulpotomy? Probably that would generate more efforts and longer operation time.

I can´t find any comments about operation time for the comprehensive treatment under anesthesia in the discussion. It is very likely that endodontic treatment is more time consuming than tooth extractions. I also would like to read a discussion about time aspect comparing tooth extraction and pulpotomy/endodontic treatment

It would have been a great advantage if you had a control group where you were more radical with tooth extractions as a comparison. Both for evaluate the operation time and mastication parameters.

Line 308 This does not represent a loss of opportunity in terms of public health as patients retain all the benefits of having kept their teeth and will be able to access conventional dentistry in the dental chair upon successfully managing their dental anxiety and/or having sufficient financial resources.

Suggestion: Develop an additional text about dental anxiety and overcoming dental phobia, which must be the final goal to achieve.

Line 324 The main study limitations are related to the short follow-up period, and to the high number of “lost to follow-up” patients, implying a five-year study to obtain the required number of subjects evaluated six months postoperatively.

Yes, I agree.

On the other hand, you write: Line 333 Excluding patients with disabilities is another limitation of this study.

No, I disagree. they do not represent the patient material you wanted to analyze. But why not include the patients I mentioned before, the healthy patients over 65 years?

Brief summery and broad comments:

Overall it is a research area of interest.

However, I wish your focus in the future will be to treat the dental anxiety instead of going further with development of dental treatment under GA.

It is a big challenge to work with patients with dental phobia and anxiety, but they can be treated if you have patience, time, resources and cost recovery.

Author Response

Specific comments:

Line 86 For patients requiring multiple dental treatments, two consecutive interventions were organized at a four to six months interval.

How did you deal with more than one intervention when you did the calculations? The chewing ability probably change over time after the first and second treatment. I can´t find any comments under the result section, missing?

Answer:

That’s right. We failed to mention this situation in the exclusion criteria. Subjects whose treatment requires two or more sessions of general anesthesia were excluded. That is now mentioned in table XXX

Line 90 Patients who demonstrated high levels of anxiety received 5 ml of midazolam orally, before going into the operating room

Concentration of midazolam is lacking!

Answer:

Midazolam concentration was 1mg/ml. We made the following correction:

Patients who demonstrated high levels of anxiety received 5 ml mg of midazolam orally, before going into the operating room.

Line 101 The next day, a nurse phoned them to check for any adverse reactions to the intervention.

I can not see any comments regarding this in the results, are there something to add?

Answer:

We agree, it would be interested to report the post-operative events. Unfortunately we did not include these details when submitting the protocol to the ethical committee and we did not collect these data for the study. For your own information, we checked that additional medications of antalgics have been prescribed for 5 of the 102 included patients. No major adverse event occurred postoperatively.

Line 107 The cohort was populated by uncooperative adult patients (between 18 and 65 years old) referred by general dentists or physicians to the Special Care Unit of the Clermont-Ferrand University Hospital after failure to be treated in the dental chair, from 01/01/2011 to 31/12/2015.

Why exclusion of patients over 65 year if they are healthy? You have 102 patients in the results (Line 195) who were excluded because of age but they got treatment under GA so the exclusion criteria were not the risk within the GA. I suggest that you follow it up in the discussion.

Answer:

We added the following paragraph in the discussion section Lines 382-388:

Excluding patients over 65 years old and those with disabilities could be discussed. Three factors have a major impact on masticatory function in elderly persons: the number of natural antagonist teeth, the quantity or/and quality of saliva and the impairment of the motor apparatus. Each of these three factors is largely correlated with ageing, and from a pragmatic point of view need to be included when considering mastication in the elderly [47]. To avoid confounding factors related to ageing, elderlies were excluded based on the 65 years old cut-off value considered by the World Health Organization to discriminate old patients. Excluding patients with disabilities is another point of discussion.

Peyron MA, Woda A, Bourdiol P, Hennequin M. Age-related changes in mastication. J Oral Rehabil. 2017 Apr;44(4):299-312. doi: 10.1111/joor.12478.

Line 285 Over the last decade, in relation to a better understanding of pulp biology, pulpotomy has been reinvestigated as a definitive treatment for mature permanent teeth and indications for pulpotomy in mature permanent teeth presenting reversible or irreversible pulpitis are now widely debated [35].

Consider a discussion about the questions below:

Did you have any postoperative complications? Non are reported

Comments about pulpotomy prognosis in adults compared to children?

Answer:

We recently published a descriptive study conducted in the same hospital unit (Linas et al 2020). Among 263 teeth treated under GA with IRM full pulpotomy and stainless steel preformed crowns, and evaluated after a median follow-up period of 24 months, 89% of the pulpotomies were effective, 7.6% were of uncertain outcome, and 3.4% were ineffective. Neither the aetiology of lesion, the tooth maturity, nor the endodontic difficulty influenced the rate of effectiveness.

Wouldn´t be better to perform more of root canal treatment directly instead of pulpotomy?

Probably that would generate more efforts and longer operation time.

Answer:

Regarding the literature, the success rate of pulpotomy are similar to those of root canal treatment, when indications are correctly assumed. Time is the adjustment’s variable for clinicians working under GA, and shortest procedures are always preferred.  You are right:  root canal treatment generates more efforts and longer operation time than pulpotomy.

I can´t find any comments about operation time for the comprehensive treatment under anesthesia in the discussion. It is very likely that endodontic treatment is more time consuming than tooth extractions. I also would like to read a discussion about time aspect comparing tooth extraction and pulpotomy/endodontic treatment

Answer:

We added the following comment in the discussion section (around line 327)

In past years, convergent results from the literature suggested that permanent vital teeth with pulpitis may be treated using full pulpotomy. Several studies have shown similar success rates for full pulpotomy compared to root canal treatment (RCT) [39]. It might be expected that full pulpotomy will become the endodontic treatment of choice for affected teeth with a vital pulp, instead of RCT. This may be particularly true when treatment is undertaken in special conditions limiting the operatory time and subsequent intervention, as under GA. A recent study reported that among 263 teeth treated with full pulpotomy under GA and evaluated after a median follow-up period of 24 months, 89% were effective, 7.6% were of uncertain outcome, and 3.4% were ineffective. Neither the aetiology of lesion, the tooth maturity, nor the endodontic difficulty influenced the effectiveness rate.

  1. Cushley S, Duncan HF, Lappin MJ, et al. Pulpotomy for mature carious teeth with symptoms of irreversible pulpitis: A systematic review. J Dent, 2019, 88, 103158. Doi: 10.1016/j.jdent.2019.06.005.

It would have been a great advantage if you had a control group where you were more radical with tooth extractions as a comparison. Both for evaluate the operation time and mastication parameters.

Answer:

Surely, including a control group would be stronger in terms of clinical research methodology, but ethically debatable. Another possibility was to dichotomize the patients in two categories, with a low and high number of extractions. That strategy would implicate to statistically control the number of restorations, as extractions and restorations have adverse effects on mastication. We did not include enough subjects in the present study to be able to categorise first the number of extractions and then the number of occlusal restorations. An additional study should be designed for such a goal.

Line 308 This does not represent a loss of opportunity in terms of public health as patients retain all the benefits of having kept their teeth and will be able to access conventional dentistry in the dental chair upon successfully managing their dental anxiety and/or having sufficient financial resources.

Suggestion: Develop an additional text about dental anxiety and overcoming dental phobia, which must be the final goal to achieve.

Answer: Line 395-407

Patients accessing dental care under general anaesthesia have very diverse biopsychosocial situations. Among them, patients with dental anxiety have special needs that require both a dental professional and a psychological therapist. The dental fear cycle was well studied by David Locker, and behavioural and cognitive therapies (BCT) are indicated to treat dental anxiety [48]. During BCT, patients need to be exposed to dental care progressively, in a positive situation without pain exposition nor negative judgment. He/she had to understand that he/she could have the control of the situation. Treating these patients under general anaesthesia avoid exposition to dental care and reinforce anxiety. However BCT programs require time and financial means that patients with dental anxiety do not have in most cases. Indeed, and particularly for patients with acute dental disease, general anaesthesia represents the only way to access dental care. It is then up to the dental professional to refer the patient to a therapist to treat his/her dental anxiety after oral rehabilitation under general anaesthesia. The benefits brought to the patient in terms of quality of life and chewing ability can be influential factors to bring back the patient in a dental chair care setting. 

48. Kankaala T, Määttä T, Tolvanen M, Lahti S, Anttonen V. Outcome of Chair-Side Dental Fear Treatment: Long-Term Follow-Up in Public Health Setting. Int J Dent. 2019  4;2019:5825067. doi: 10.1155/2019/5825067

Line 324 The main study limitations are related to the short follow-up period, and to the high number of “lost to follow-up” patients, implying a five-year study to obtain the required number of subjects evaluated six months postoperatively.

Yes, I agree.

On the other hand, you write: Line 333 Excluding patients with disabilities is another limitation of this study.

No, I disagree. they do not represent the patient material you wanted to analyze. But why not include the patients I mentioned before, the healthy patients over 65 years?

Answer:

The answer was given above .